# Causal role of metabolites in non-small cell lung cancer: Mendelian randomization (MR) study

Qian Li[1]*, Zedong Wei[2], Yonglun Zhang[2], Chongqing Zheng[2]

**1** Department of Cardiac Surgery, The First Affiliated Hospital of Kunming Medical University, Kunming, 650000, China, **2** Department of Thoracic Surgery, Qianjiang Central Hospital, Qianjiang, 433100, China

* liqian20210703@icloud.com

## Abstract

On a global scale, lung cancer(LC) is the most commonly occurring form of cancer. None-theless, the process of screening and detecting it in its early stages presents significant challenges. Earlier research endeavors have recognized metabolites as potentially reliable biomarkers for LC. However, the majority of these studies have been limited in scope, featuring inconsistencies in terms of the relationships and levels of association observed.Moreover, there has been a lack of consistency in the types of biological samples utilized in previous studies. Therefore, the main objective of our research was to explore the correlation between metabolites and Non-small cell lung cancer (NSCLC).Thorough two-sample Mendelian randomization (TSMR) analysis, we investigated potential cause-and-effect relationships between 1400 metabolites and the risk of NSCLC.The analysis of TSMR revealed a significant causal impact of 61 metabolites on NSCLC.To ensure the reliability and validity of our findings, we perform FDR correction for P-values by Benjaminiand Hochberg(BH) method, Our results indicate that Oleate/vaccenate (18:1) levels and Caffeine to para-xanthine ratio may be causally associated with an increased risk of NSCLC [Oleate/vaccenate(18:1)levels: OR = 1.171,95%CI: 1.085–1.265, FDR = 0.036; Caffeine to paraxanthine ratio: OR = 1.386, 95%CI:1.191–1.612,FDR = 0.032].

## Introduction

LC is a prevalent form of cancer, accounting for 12.9% of global cancer-related fatalities and causing approximately 1.3 million deaths annually [1–3]. Approximately 80.0 to 85.0% of LC cases are categorized as NSCLC [3–5]. The late diagnosis of LC poses a significant challenge, resulting in a short median survival of about 18 months and an overall 5-year survival rate ranging from 15% to 21% depending on gender [6,7]. Sadly, as many as 75% of individuals diagnosed with NSCLC have already reached an advanced stage or developed metastases at the time of diagnosis, leading to a dismal 5-year survival rate below 5% [1]. Despite extensive research conducted to investigate LC, including studying the molecular mechanism, cytological behavior, and biological process, our available treatments for LC remain inadequate. Previous researches have shown that metabolites are crucial factor affecting LC. In LC, researchers

**Data Availability Statement:** The GWAS data for non-small cell lung cancer comes from the database (http://gwas.mrcieu.ac.uk), and can be downloaded via the GWAS ID "finn-b-C3_LUNG_NONSMALL". The GWAS data for 1,400

metabolites comes from the database (https://www.ebi.ac.uk/gwas/), and can be downloaded using the European GWASs registration number: GCST90199621-90201020. The results of the Mendelian randomization analysis related to the article can be found in the supplementary files.

**Funding:** The author(s) received no specific funding for this work.

**Competing interests:** The authors declare that they have no competing interests.

have conducted metabolomic studies in blood and urine samples to discover metabolite biomarkers for diagnosing lung cancer [8–14], characterizing tumors [15,16], and monitoring disease progression [8,11]. However, there is a lack of a conclusive and all-encompassing summary regarding the causal relationship between metabolites and LC, and this research aims to address this gap.

Metabolites refer to the substances produced and utilized in the biological metabolism of both host organisms and microorganisms and xenobiotics [17]. Metabolomics is a widely employed technique in which the analysis of metabolites present in biofluids, cells, and tissues is carried out. This approach is commonly utilized for the identification of biomarkers [18]. By employing metabolites obtained from human studies, it becomes possible to establish links between diseases and metabolic pathways [19]. Due to the high sensitivity of metabolomics, even small changes in biological behavior can be identified and used to gain understanding of the mechanisms behind different physiological conditions, abnormal processes, and diseases [17]. Despite the long history of metabolite identification and validation, a major obstacle in biomarker validation remains overcoming the variations in metabolite levels among individuals, caused by differences in genetic factors and environmental exposures [20,21]. Identifying the metabolites and understanding their biological roles is a crucial step in cancer research. Numerous studies have demonstrated the immense potential of metabolomics in this field [22,23].

MR is a technique that utilizes genetic variants as instrumental variables (IVs) to assess the causal effects between exposure and outcome. It is a widely applicable method built upon Genome-Wide Association Studies (GWAS) summary data [24]. To put it differently, the genetic variants found in GWAS can serve as a substitute for a randomized controlled trial to establish a cause-and-effect relationship between different factors [22,23]. Additionally, GWAS has been expanded to include metabolic profiles. In this research, an extensive TSMR analysis was conducted to ascertain the causal link between metabolites and NSCLC.

## Materials and methods

### Study design

Using a TSMR study, we conducted a comprehensive analysis to establish the causal impacts of 1400 metabolites on the susceptibility to NSCLC. In the MR study, three basic assumptions need to be adhered to [25]. Firstly, the genetic variants must display a strong correlation with the exposure factor. Secondly, they must be independent of any potential factors that may confound the results. Lastly, it is essential that these genetic instruments impact the outcome solely through the exposure.

### GWAS data sources for NSCLC and metabolites

We performed a search using the GWAS summary data(http://gwas.mrcieu.ac.uk), which includes a comprehensive collection of summary statistics from multiple GWAS. For our analysis, we utilized the openly available summary statistic datasets obtained from a GWAS conducted on individuals with European ancestry diagnosed with NSCLC (total n = 217165; ncase = 1627, ncontrol = 217,165).Various metabolites GWAS summary statistics were deposited to GWAS Catalog (https://www.ebi.ac.uk/gwas/). Accession numbers for European GWASs: GCST90199621-90201020. These statistics cover a total of 1400 metabolites [26].

### Selection IVs

Based on our study, we established a significance level of $1×10−5$ for the independent variables each metabolite. In order to obtain independent IVs, we employed clumping technique by

using a linkage disequilibrium (LD) reference panel sourced from the 1000 Genomes Project. Clumping was performed based on a criterion of R2 < 0.001 within a 1,000-kb distance [27]. For NSCLC, when we adjusted the significance level to 5×10−8,there is only one significant instrumental variable. therefore, we adjusted the significance level to 1×10−5 for NSCLC.Furthermore, in order to mitigate any potential bias caused by weak instruments, we only included IVs with F statistics greater than 10, which we identified as strong instruments, for the subsequent analysis.We harmonized tthe exposure and outcome single nucleotide polymorphism(SNPs) so that the effect estimates were consistent for the same effect allele. We removed palindromic SNPs with intermediate effect allele frequencies (EAFs > 0.42) or SNPs with incompatible alleles [28].

## Statistical analysis

The data analysis in the study was carried out using R software (version 4.2.1). The TwoSampleMR package (version 0.5.6) and MRPRESSO package (version 1.0) were utilized for the analysis [29]. As our primary analysis, we used the random-effects inverse variance weighted (IVW), weighted median (WM), MREgger(ME),Simple mode(SM) and Weighted mode(WM) methods,providing an estimate of the effect of the exposure on the outcome when MR assumptions are valid.We applied Cochran's Q test (p<0.05) to estimate residual heterogeneity for the IVW model and the MR-Egger intercept test (p < 0.05) to indicate potential pleiotropy on causal estimates. In order to eliminate the influence of horizontal pleiotropy, the researchers employed a widely-used approach known as MR-Egger. If the intercept term of MR-Egger is found to be statistically significant, it suggests the existence of horizontal multiplicity [30]. Furthermore, the MR pleiotropy residual sum and outlier (MR-PRESSO) method, which is a robust approach, was employed to eliminate potential horizontal pleiotropic outliers that could have had a significant impact on the estimation outcomes within the MR-PRESSO package [31]. In addition, we correction for P-values (FDR<0.05) with BH method make the results more rigorous and reliable.

# Results

## Causal estimates between metabolites and NSCLC

To explore the causal effects of 1400 metabolites on NSCLC, TSMR analysis was performed, and the IVW method was used as the main analysis (S1 Table). For sensitivity analysis, we utilized the Cochranwith a significance level of p < 0.05) to assess heterogeneity in the IVW model. Additionally, we performed the MR-Egger intercept test (with a significance level of p < 0.05) to evaluate the potential impact of pleiotropy on the causal estimates (S2 and S3 Tables). Finally, a leave-one-out (LOO) analysis was conducted to determine if a single nucleotide polymorphism (SNP) could influence the accuracy of the causal estimate. 61 metabolites were found to be potentially significant, with a significance level of P<0.05.After correction for P-values (FDR<0.05) with BH method, we detected the significant effect of two metabolites(Oleate/vaccenate (18:1) levels and Caffeine to paraxanthine ratio) on NSCLC (S4 Table). Oleate/vaccenate (18:1) levels and Caffeine to paraxanthine ratio is a risk factor for NSCLC (Fig 1). Our study displays the relationship between the two metabolites and the risk of NSCLC, indicating no heterogeneity among the IVs with a P-value of 0.679 and 0.617 (S2 and S4 Tables). The Forest plots in Fig 2 depict the impact of SNP effect sizes on Oleate/vaccenate (18:1) levels and Caffeine to paraxanthine ratio. The leave-one-out analysis does not uncover any statistically significant outliers or influential data points (Fig 3). Our results support genetic relationship between the two metabolites and NSCLC (Fig 4). Notably, Significantly, our findings have revealed a robust cause-and-effect relationship between a heightened

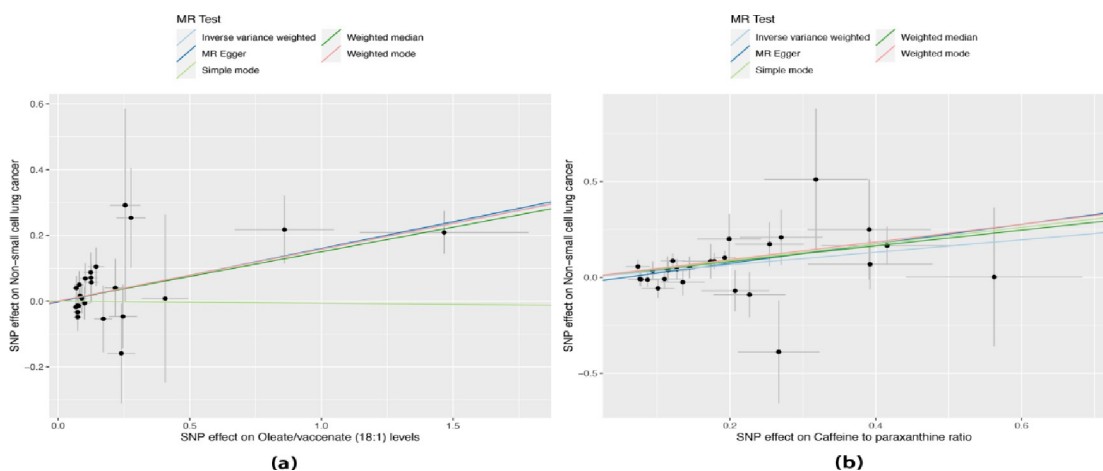

**Fig 1. Scatterplot of genetic association Oleate/vaccenate (18:1) levels and Caffeine to paraxanthine ratio With NSCLC.** (a) Oleate/vaccenate (18:1) levels with NSCLC and (b) Caffeine to paraxanthine ratio With NSCLC.

susceptibility to NSCLC and elevated levels of the two metabolites, as evidenced by rigorous IVW analysis[Oleate/vaccenate (18:1) levels:odds ratio (OR) = 1.172, 95% confidence interval (CI) = 1.085–1.265, P<0.001; Caffeine to paraxanthine ratio:OR = 1.386, CI = 1.192–1.612, P<0.001], Similar positive trends with weighted median and weight mode methods were observed (Fig 4, S1 Table).

Then we perform reverse MR analysis, we investigated the relationship between NSCLC as the exposure and the two metabolites(Oleate/vaccenate (18:1) levels and Caffeine to para-xanthine ratio) as the outcome. We identified 16 SNPs that were found to be associated with these variables. The results of our analysis revealed no significant correlations between NSCLC and the two metabolites.as shown by the IVW (P = 0.618 and P = 0.904),as depicted in Fig 5.

## Discussion

Using extensive genetic data that is publicly accessible, we conducted an analysis to investigate the causal connections between 1400 metabolites and NSCLC. As far as we know, this is the initial MR study to examine the causal relationship between numerous metabolites and

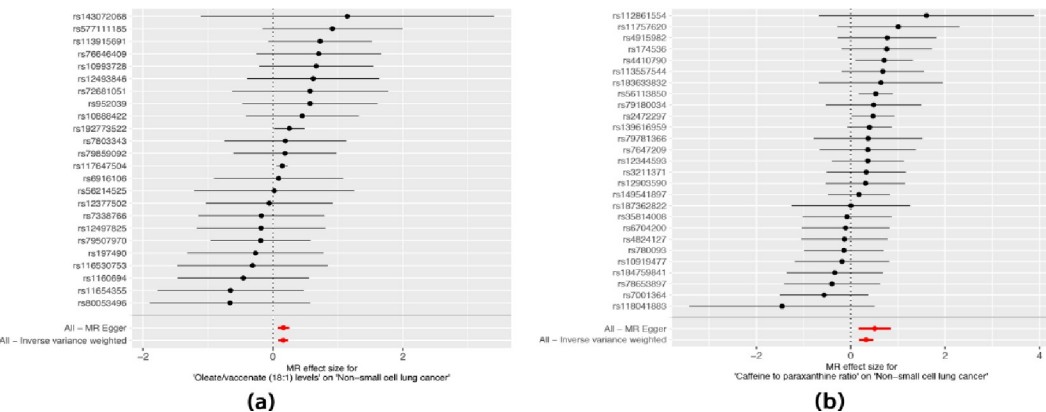

**Fig 2.** Forest plots of causal efects of Oleate/vaccenate (18:1) levels-associated SNPs on NSCLC (a) and Caffeine to paraxanthine ratio-associated SNPs on NSCLC (b).

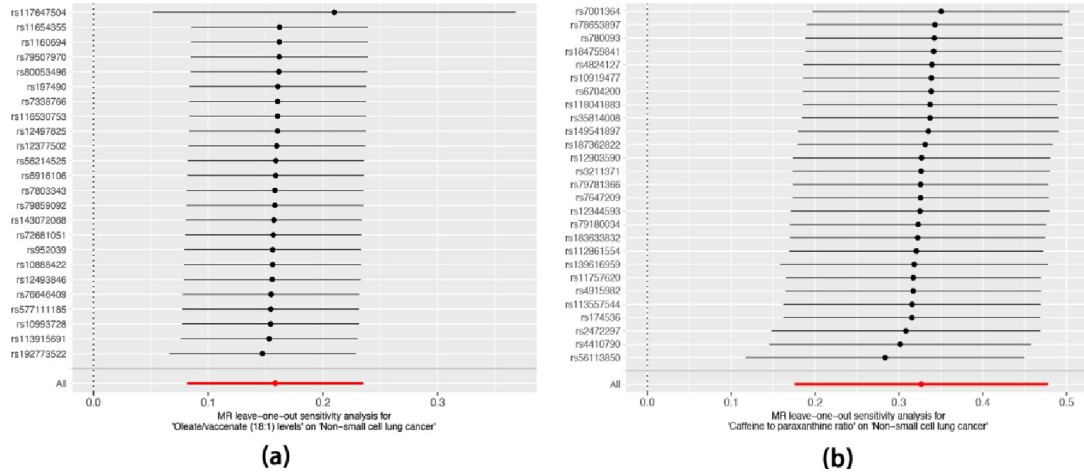

**Fig 3.** Leave-one-out analysis of SNPs associated with Oleate/vaccenate (18:1) levels and NSCLC(a) and Caffeine to paraxanthine ratio and NSCLC(b).

NSCLC. In this study, 61 metabolites was found to have causal effects on NSCLC(P<0.05), and two metabolites(Oleate/vaccenate (18:1) levels and Caffeine to paraxanthine ratio) had significant causal effects on NSCLC(FDR<0.05). However, the reverse MR analysis representation NSCLC had no significant causal effects on the two metabolites.

Our study found that the risk of NSCLC increase with an increase in the levels of two metabolites(Oleate/vaccenate (18:1) levels and Caffeine to paraxanthine ratio). These two metabolites have not been linked to lung cancer in previous studies. Oleate/vaccenate (18:1) levels are monounsaturated fatty acids(MUFA). It is an important component of lipid synthesis. Lipids play a crucial role in cellular growth and division, as well as in the formation of new cell membranes, energy production, and signaling processes. Therefore, it is not surprising that cancer cells necessitate greater amounts of lipids to sustain their unregulated proliferation [32,33]. To meet this condition, cancer cells boost their lipid production through de novo lipogenesis or increase their uptake of lipids, in order to make use of lipids obtained from their surrounding microenvironment. Complex lipids are synthesized using fatty acids as their building blocks. The properties of these fatty acids, such as their length and degree of unsaturation, play a significant role in determining the properties of the resulting lipids. In general, fatty acids can be categorized into three primary groups: Saturated fatty acids(SFA), MUFA, and polyunsaturated fatty acids(PUFA) [34]. Cancer has been associated with oleic acid (OA), the most prevalent MUFA found in the bloodstream, according to several research studies. OA has been proven to be consumed and utilized by metastatic breast cancer and prostate cancer cells in culture, indicating its association with the development and progression of cancer [35–37]. These studies emphasize the significance of MUFAs and OA in relation to cancer, as

| exposure | nsnp | method | pval | | OR(95% CI) |
|---|---|---|---|---|---|
| Oleate/vaccenate (18:1) levels | 24 | Weighted median | 0.010 | | 1.162 (1.036 to 1.303) |
| | 24 | Inverse variance weighted | <0.001 | | 1.172 (1.085 to 1.265) |
| | 24 | Weighted mode | 0.006 | | 1.170 (1.057 to 1.296) |
| Caffeine to paraxanthine ratio | 27 | Weighted median | <0.001 | | 1.508 (1.219 to 1.865) |
| | 27 | Inverse variance weighted | <0.001 | | 1.386 (1.192 to 1.612) |
| | 27 | Weighted mode | 0.001 | | 1.587 (1.236 to 2.036) |

**Fig 4. MR estimates for the relationship Oleate/vaccenate (18:1) levels and Caffeine to paraxanthine ratio with NSCLC.**

| outcome | nsnp | method | pval | | OR(95% CI) |
|---|---|---|---|---|---|
| Oleate/vaccenate (18:1) levels | 16 | Weighted median | 0.497 | | 1.021 (0.962 to 1.082) |
| | 16 | Inverse variance weighted | 0.618 | | 1.010 (0.970 to 1.053) |
| | 16 | Weighted mode | 0.296 | | 0.935 (0.828 to 1.056) |
| Caffeine to paraxanthine ratio | 16 | Weighted median | 0.153 | | 1.043 (0.984 to 1.105) |
| | 16 | Inverse variance weighted | 0.904 | | 0.997 (0.945 to 1.052) |
| | 16 | Weighted mode | 0.200 | | 1.048 (0.979 to 1.122) |

**Fig 5. Reverse MR estimates for the relationship NSCLC with Oleate/vaccenate (18:1) levels and Caffeine to paraxanthine ratio.**

cancer cells specifically amplify MUFA levels and utilization. Stearoyl CoA desaturase (SCD1) is a vital enzyme in the production of fresh lipids. It is found to be excessively expressed in melanoma, colorectal cancer, and clear-cell renal carcinoma [38–40]. Moreover, elevated levels of SCD1 have been linked to the progression of lung cancer and unfavorable prognoses in patients [41]. The previously unexplored n-7 product of SCD1, known as cis-vaccenic acid, has been revealed to play a significant role in regulating the growth of prostate cancer cells and oxidative stress [42]. This is in spite of numerous studies that have established a connection between Oleate/vaccenate (18:1) levels and cancer, but they are rarely associated with lung cancer. Our study provides new evidence for future research.

Caffeine is completely absorbed, rapidly distributed and extensively metabolized in the liver to three major metabolites: paraxanthine (1,7 dimethylxanthine), theophylline (1,3 dimethylxanthine), and theobromine (3,7 dimethylxanthine) which undergo further metabolism to several xanthines and uric acids [43]. The metabolism of caffeine is primarily carried out by CYP1A2, with more than 95% of the substance being metabolized through this enzyme. The variability in the activity of CYP1A2 among individuals is believed to strongly correlate with the differences observed in caffeine metabolism [44–46]. The paraxanthine to caffeine ratio provides valuable information as paraxanthine is formed solely through the catalytic action of CYP1A2 [47]. This step accounts for approximately 80% of caffeine metabolism. There is a scarcity of reports exploring the association between caffeine and LC. It is possible that caffeine could hinder the repair of DNA, thereby increasing the vulnerability to lung adenocarcinoma in women with rapid CYP1A2 activity and slow N-acetyltransferase2(NAT2) [48,49]. No link between Caffeine to paraxanthine ratio and lung cancer risk has been reported.Our research may provide clues for future exploration.

This study conducted two-sample MR analysis based on the published results of large GWAS cohorts,with a large sample size of 1400 metabolites,so it has high statistical efficiency. The conclusions of this study are based on genetic instrumental variables, and causal inference is made using a variety of MR analysis methods. The results are robust and were not confounded by horizontal pleiotropy and other factors. Our study has limitations. First,the study was based on a European database, so the conclusion cannot be extended to other ethnic groups. Second, we used strict threshold to evaluate the results, which may decrease some true positives.Finally, our study was limited to non-small cell lung cancer, and the relationship between metabolites and risk of small cell lung cancer needs to be explored in future studies.

## Conclusions

In summary, our comprehensive bidirectional MR analysis has established the causal associations between certain metabolites and NSCLC. This sheds light on the intricate interactions between metabolites and NSCLC.

Additionally, our research offers researchers a fresh opportunity to delve into the biological mechanisms of NSCLC, potentially paving the way for earlier intervention and treatment options. By building upon the existing knowledge in metabolomics of NSCLC, our findings present crucial hints for preventing the onset of NSCLC.

## Supporting information

**S1 Table. Results of the causal effect of metabolites on NSCLC.**
(XLSX)

**S2 Table. Sensitivity analysis results of causal effects of metabolites on NSCLC.**
(XLSX)

**S3 Table. Pleiotropy analysis results of causal effects of metabolites on NSCLC.**
(XLSX)

**S4 Table. 61 metabolites associated with NSCLC.**
(XLSX)

## Author Contributions

**Conceptualization:** Qian Li.

**Data curation:** Qian Li, Zedong Wei.

**Project administration:** Zedong Wei.

**Supervision:** Chongqing Zheng.

**Validation:** Yonglun Zhang.

**Visualization:** Zedong Wei.

**Writing – original draft:** Qian Li.

**Writing – review & editing:** Yonglun Zhang, Chongqing Zheng.

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
