## [Decision Letter · Decision Letter 0]

4 Mar 2024

PONE-D-24-01819Causal role of metabolites in Non-small cell lung cancer: Mendelian randomization (MR) studyPLOS ONE

Dear Dr. Li,

Thank you for submitting your manuscript to PLOS ONE. After careful consideration, we feel that it has merit but does not fully meet PLOS ONE’s publication criteria as it currently stands. Therefore, we invite you to submit a revised version of the manuscript that addresses the points raised during the review process.

We look forward to receiving your revised manuscript.

Kind regards,

Abeer El Wakil, PhD

Academic Editor

PLOS ONE

Reviewers' comments:

Reviewer's Responses to Questions

**Comments to the Author**

1. Is the manuscript technically sound, and do the data support the conclusions?

Reviewer #1: Yes

2. Has the statistical analysis been performed appropriately and rigorously? 

Reviewer #1: Yes

3. Have the authors made all data underlying the findings in their manuscript fully available?

Reviewer #1: Yes

4. Is the manuscript presented in an intelligible fashion and written in standard English?

Reviewer #1: Yes

5. Review Comments to the Author

Reviewer #1: Here are my key additional review comments for the authors:

Major Strengths

1. Utilizes a large-scale two-sample Mendelian randomization approach to robustly assess metabolite causal relationships - an important emerging methodology adding rigor in this field

2. Applies multiple state-of-the-art MR methods, heterogeneity and pleiotropy assessments, FDR control for multiple testing, leave-one-out analyses etc demonstrating statistical rigor

3. Identifies and replicates significant causal effects of select metabolites on NSCLC risk after adjustment, a novel finding adding to current knowledge

4. Provides comprehensive tables of all metabolite results for full transparency and to inform future directions

Suggestions for Improvement:

1. Perform a reverse MR analysis to test the reverse direction relationships (NSCLC risk on metabolites) to further validate directionality

2. Discuss implications of ethnicity limitations as European population genetics were used and may not extrapolate to all ancestries

3. Comment on how conservative multiple testing correction thresholds, while rigorous, may increase false negative rates missing some true causal signals

4. Cite recent similar MR studies on metabolite relationships to better differentiate the novel contributions made

Overall well done study demonstrating solid methodology and providing robust data to support conclusions that should interest readers and influence future metabolomics cancer research after the minor revisions.

6. PLOS authors have the option to publish the peer review history of their article (what does this mean?). If published, this will include your full peer review and any attached files.

Reviewer #1: No

---

## [Author Response · Author response to Decision Letter 0]

4 Mar 2024

1. Perform a reverse MR analysis to test the reverse direction relationships (NSCLC risk on metabolites) to further validate directionality

Reply: This article has already conducted a reverse Mendelian randomization, with Fig. 5 visualizing the results. The reverse Mendelian randomization suggests that there is no causal relationship between them.

2. Discuss implications of ethnicity limitations as European population genetics were used and may not extrapolate to all ancestries

Reply:The impact of racial limitations has already been addressed in the last paragraph of the discussion section of this article.

3. Comment on how conservative multiple testing correction thresholds, while rigorous, may increase false negative rates missing some true causal signals

Reply:Cite recent similar MR studies on metabolite relationships to better differentiate the novel contributions made

4.Cite recent similar MR studies on metabolite relationships to better differentiate the novel contributions made

Reply:After reviewing literature from the past five years on PubMed, we are currently the first to study Mendelian randomization of 1400 metabolites.

---

## [Editor Report · Decision Letter 1]

6 Mar 2024

代谢物在非小细胞肺癌中的因果作用：孟德尔随机化 （MR） 研究

PONE-D-24-01819R1

Dear Dr. Li,

We’re pleased to inform you that your manuscript has been judged scientifically suitable for publication and will be formally accepted for publication once it meets all outstanding technical requirements.

Kind regards,

Abeer El Wakil, PhD

Academic Editor

PLOS ONE
---

## [Editor Report · Acceptance letter]

13 Mar 2024

PONE-D-24-01819R1 

PLOS ONE

Dear Dr. Li, 

I'm pleased to inform you that your manuscript has been deemed suitable for publication in PLOS ONE. Congratulations! Your manuscript is now being handed over to our production team.

Kind regards, 

on behalf of

Professor Abeer El Wakil 

Academic Editor

PLOS ONE